# Whole-Genome Sequencing-Based Characteristics in Extended-Spectrum Beta-Lactamase-Producing *Escherichia coli* Isolated from Retail Meats in Korea

**DOI:** 10.3390/microorganisms8040508

**Published:** 2020-04-02

**Authors:** Seokhwan Kim, Hansol Kim, Yonghoon Kim, Migyeong Kim, Hyosun Kwak, Sangryeol Ryu

**Affiliations:** 1Division of Food Microbiology, National Institute of Food and Drug Safety Evaluation, Cheongju 28159, Korea; myksh@korea.kr (S.K.); hskmfds@korea.kr (H.K.); washout71@korea.kr (Y.K.); angelmg@korea.kr (M.K.); 2Department of Food and Animal Biotechnology, Seoul National University, Seoul 08826, Korea; 3Department of Agricultural Biotechnology, Seoul National University, Seoul 08826, Korea

**Keywords:** *Escherichia coli*, extended-spectrum beta-lactamase, ESBL, retail meat, whole-genome sequencing

## Abstract

The spread of extended-spectrum beta-lactamase-producing *Escherichia coli* (ESBL-EC) has posed a critical health risk to both humans and animals, because resistance to beta-lactam antibiotics makes treatment for commonly infectious diseases more complicated. In this study, we report the prevalence and genetic characteristics of ESBL-ECs isolated from retail meat samples in Korea. A total of 1205 *E. coli* strains were isolated from 3234 raw meat samples, purchased from nationwide retail stores between 2015 and 2018. Antimicrobial susceptibility testing was performed for all isolates by a broth microdilution method, and the ESBL phenotype was determined according to the Clinical and Laboratory Standards Institute (CLSI) confirmatory method. All ESBL-EC isolates (*n* = 29) were subjected to whole-genome sequencing (WGS). The antimicrobial resistance genes, plasmid incompatibility types, *E. coli* phylogroups, and phylogenetic relations were investigated based on the WGS data. The prevalence of ESBL-ECs in chicken was significantly higher than that in other meat samples. The results in this study demonstrate that clonally diverse ESBL-ECs with a multidrug resistance phenotype were distributed nationwide, although their prevalence from retail meat was 0.9%. The dissemination of ESBL-ECs from retail meat poses a potential risk to consumers and food-handlers, suggesting that the continuous surveillance of ESBL-ECs in retail meat should be conducted at the national level.

## 1. Introduction

*Escherichia coli* is a ubiquitous bacterium residing in the intestinal tract of humans and animals, environment, and food. Although most *E. coli* strains are harmless commensal bacteria, some strains, which harbor virulence factors, can cause various infections, such as diarrhea, hemorrhagic colitis, urinary tract infection, and meningitis [1,2,3]. *E. coli* can also serve as an important reservoir of antimicrobial resistance (AMR) genes, that may be transferred to human pathogenic bacteria [4,5,6]. Therefore, the spread of antimicrobial-resistant *E. coli*, especially extended-spectrum beta-lactamase (ESBL)-producing *E. coli* (ESBL-ECs)*,* has become a threat to human as well as animal health worldwide [7].

ESBLs are enzymes that confer resistance to most beta-lactams, such as penicillins and cephalosporins, except for cephamycin or carbapenem, but these enzymes are inhibited by clavulanate [8]. The resistance to beta-lactams, one of the most widely used antibiotics, makes treatment for common infectious diseases caused by ESBL-ECs more complicated, as it involves hospitalization and intravenous carbapenem administration, instead of taking oral antibiotics at home [9,10,11]. As some ESBL genes are located on mobile elements such as plasmids and may be easily transferred to various bacterial species [12,13], the prevalence of ESBL-producing isolates from humans, livestock, and even food is rapidly increasing worldwide [14,15]. Some previous studies have suggested that ESBL genes can be disseminated through the food chain [16,17,18]. Meat contaminated by antimicrobial-resistant bacteria can act as a reservoir of such bacteria, and resistance determinants may be transferred to humans [18,19]. The selection pressure due to the ongoing overuse and misuse of antimicrobial agents possibly accelerates the emergence of antimicrobial-resistant bacteria [20,21].

The Korean government has monitored the antimicrobial susceptibilities of zoonotic bacteria from foods such as retail meat to medically important antimicrobials including beta-lactam antibiotics. Since whole-genome sequencing (WGS) has become affordable and facilitates the acquisition of useful information regarding multiple AMR genes, genomic mutations, and higher-resolved microbial typing from a single assay, some countries, including the USA, have already conducted WGS-based AMR surveillance [22]. A number of studies have reported the prevalence and characteristics of ESBL-EC from humans and food-producing animals in Korea because of the importance of such ESBL-ECs from a public health perspective [23,24,25,26,27,28,29]. However, few studies have reported the prevalence and characteristics of ESBL-ECs in retail meat in Korea [30,31,32]. Therefore, this study aimed to report the prevalence and AMR-related characteristics of ESBL-ECs present in retail meat, including beef, pork, and chicken, collected through the national surveillance program between 2015 and 2018. This information will help to characterize the molecular epidemiology of ESBL-ECs related to retail meat in Korea.

## 2. Materials and Methods

### 2.1. Sample Collection and Bacteria Isolation

In total, 3234 meat samples, including beef (*n* = 1,290), pork (*n* = 1,126), and chicken (*n* = 818), were purchased at approximately 100 grocery stores spread across all the provinces of South Korea between 2015 and 2018. Overall, average ~800 raw meat samples were purchased per year. Domestic meat samples were from 43 beef production companies, 32 pork production companies, and 18 chicken production companies; these companies had high market shares. Among the imported meat samples, beef samples were from 5 countries, pork from 14 countries, and chicken from 4 countries. The meat samples were kept on ice during transportation from the grocery stores to the laboratory. Twenty-five grams of each meat sample was homogenized with 225 mL EC broth (Difco, MI, USA) using a stomacher. The homogenized samples were incubated under aerobic conditions at 37 °C for 24 h. An aliquot of each sample was streaked onto selective medium, the Eosin Methylene Blue agar (Oxoid, Cambridge, UK), and incubated at 37 °C for 24 h. Typical *E. coli* colonies (green metallic sheen) were sub-cultured on nutrient agar (Difco) and confirmed using a Vitek 2 Compact microbial identification system (bioMérieux, France) or Vitek MS (bioMérieux) by following the manufacturer’s instructions. One typical and well-isolated *E. coli* strain per meat sample was selected. If no typical growth was observed, the sample was treated as a negative sample and was discarded. A total of 1205 *E. coli* strains were isolated from raw meat samples. All isolates were stored at −80 °C in Tryptic Soy Broth (Difco), mixed with 15% glycerol.

### 2.2. Antimicrobial Susceptibility Testing and Confirmation of ESBL-ECs

All selected strains of *E. coli* (*n* = 1,205) were subjected to antimicrobial susceptibility testing using the following antimicrobials: amoxicillin/clavulanic acid (AmC), ampicillin (AMP), cefoxitin (FOX), ceftiofur (CTF), ceftazidime (CAZ), cefepime (FEP), chloramphenicol (CHL), ciprofloxacin (CIP), colistin (COL), gentamicin (GEN), meropenem (MEM), nalidixic acid (NAL), streptomycin (STR), tetracycline (TET), and trimethoprim/sulfamethoxazole (SXT). The minimum inhibitory concentrations (MICs) of these antimicrobials were determined using a broth-dilution method that involved a commercially available Sensititre plate KRNV4F (Trek Diagnostic Systems, Cleveland, OH, USA) and by following the manufacturer’s instructions. *E. coli* ATCC 25922 was used as a reference strain. Susceptibility results in the form of MICs were interpreted by referring to the Clinical and Laboratory Standards Institute (CLSI) guidelines [33], European Committee on Antimicrobial Susceptibility Testing (EUCAST) guidelines [34], and National Antimicrobial Resistance Monitoring System [35] (Appendix A). The strains (*n* = 120) resistant to the third-generation cephalosporins (ceftiofur or ceftazidime) were tested for the ESBL phenotype, which was determined using the CLSI confirmatory broth microdilution test, that involves ceftazidime and cefotaxime with and without clavulanic acid [33].

### 2.3. Whole-Genome Sequencing and Phylogenetic Analysis

All isolates with an ESBL phenotype (*n* = 29) were subjected to whole-genome sequencing (WGS). Total bacterial DNA was extracted using an UltraClean microbial DNA isolation kit (MO BIO Laboratories Inc., Carlsbad, CA, USA), by following the manufacturer’s instructions. Sequencing was performed at Senigen Inc. (Seoul, Korea) using the Illumina MiSeq desktop sequencer (Illumina Inc., San Diego, CA, USA), with paired-end reads of length 300 bp. A de novo assembly was performed using SPAdes genome assembler version 3.13.0 [36]. Contigs of less than 200 bp in length and 5× in sequencing depth were removed from analysis. The number of assembled contigs ranged between 41 and 202, with an average sequencing depth of 210×. These assemblies were annotated with Prokka [37] and the output was used for the pan-genome pipeline using Roary [38] to construct the core-genome of 29 ESBL-EC isolates. Roary parameters were set to default (minimum blastp identity 95% and threshold of isolates required to define a core gene 99%). Genes were classified as “core” and “soft core” if they were identified in at least 99% of the isolates and 99%–95% of the isolates, respectively. All genes present in <95% of the isolates were classified as “accessory”. The curve-fitting of the pan-genome growth was performed using a power law regression based on Heap’s law [39,40,41] as follows: $\;y=Ax^{\gamma }+B$. The fitting was conducted using PanGP [42] to fit the power law regression, where *y* and *x* are pan-genome size and number of the genomes, respectively. Furthermore, γ is an empirical parameter for estimating whether a pan-genome is open or closed [39,40].

An alignment of polymorphic sites in the core genome alignment were generated using a SNP-Sites tool [43] (https://github.com/sanger-pathogens/snp-sites). This single-nucleotide polymorphism (SNP) alignment was used to construct a maximum likelihood (ML) phylogenetic tree, using RAxML version 8.0.0 [44] under the general time reversible (GTR) substitution model with a Gamma rate of correction heterogeneity. This core-genome SNP alignment was also used to cluster the isolates into unique subpopulations or sequence clusters using the Bayesian analysis of population structure (hierBAPS) [45]. Phylogenetic trees were visualized using FigTree (https://github.com/rambaut/figtree/releases) and Phandango [46].

### 2.4. Nucleotide Sequence Accession Numbers

The whole-genome sequencing data reported in this study have been deposited at GenBank under the BioProject PRJNA599028 as the following accession numbers: WVUS00000000 (EC2015_85), WVUT00000000 (EC2016_8), WVUU00000000 (EC2016_31), WVUV00000000 (EC2016_174), WVUW00000000 (EC2016_I10), WVUX00000000 (EC2016_I174), WVUY00000000 (EC2016_I177), WVUZ00000000 (EC2017_2), WVVA00000000 (EC2017_136), WVVB00000000 (EC2017_202), WVVC00000000 (EC2017_203), WVVD00000000 (EC2017_240), WVVE00000000 (EC2017_617), WVVF00000000 (EC2017_303), WVVG00000000 (EC2017_575), WVVH00000000 (EC2017_I80), WVVI00000000 (EC2017_I216), WVVJ00000000 (EC2017_I306), WVVK00000000 (EC2017_I318), WVVL00000000 (EC2017_I327), WVVM00000000 (EC2018_100), WVVN00000000 (EC2018_102), WVVO00000000 (EC2018_273), WVVP00000000 (EC2018_311), WVVQ00000000 (EC2018_521), WVVR00000000 (EC2018_526), WVVS00000000 (EC2018_I302), WVVT00000000 (EC2018_I235), and WVVU00000000 (EC2018_I73).

### 2.5. In silico Molecular Typing and Characterization

In silico *E. coli* phylotyping was performed using ClermonTyping [47], and *E. coli* isolates were assigned to phylogroups A, B1, B2, C, D, E, and F. In silico plasmid typing was done by searching for plasmid incompatibility groups in PlasmidFinder 2.1 database [48], available on the Center for Genomic Epidemiology (CGE) website (http://www.genomicepidemiology.org). AMR genes were also identified using the ResFinder 3.2 database [49] on the CGE websites. Presence of a gene in an isolate was confirmed if its assembled genome sequence had more than 95% nucleotide identity match with a gene in the database, and a coverage of 100% of the length of the database match.

### 2.6. Multilocus Sequence Typing

ESBL-EC isolates were subjected to multilocus sequence typing (MLST), using seven housekeeping genes (*adk*, *fumC*, *gyrB*, *icd*, *mdh*, *purA*, *recA*), as previously described [50]. PCR amplification was performed using a thermal cycler 3500XL (Applied Biosystems, Singapore), under the following condition: 25 cycles of 96 °C for 10 s, 50 °C for 5 s, and 60 °C for 4 min. The internal fragments of all loci were sequenced, and the corresponding sequence types of the isolates were designated according to the *E. coli* MLST database (http://mlst.warwick.ac.uk/mlst/dbs/Ecoli). We also conducted in silico MLST using MLST 2.0 [51] on the CGE website to cross-check the sequence types (STs).

### 2.7. Statistical Data Analysis

The 95% confidence intervals (CI) of proportions were calculated with EPi tools (http://epitools.ausvet.com.au) using the binomial exact method. Statistical significance of differences between proportions was evaluated by Chi-square (χ^2^) test. Means of pairwise SNP differences for identified clusters were compared using a one-way analysis of variance (ANOVA) with sigmaplot 12.5 (Systat Software Inc., San Jose, CA, USA).

## 3. Results

### 3.1. Prevalence of ESBL-EC from Retail Raw Meat Samples

Totally, we isolated 1205 *E. coli* strains from 3234 retail meat samples, purchased from nationwide grocery stores in Korea between 2015 and 2018. These retail meat samples comprised mainly beef cuts, pork cuts, and chicken cuts. *E. coli* was present in 37.2% of the tested samples. Out of 1205 *E. coli* strains isolated from meat samples, 120 strains were resistant to third-generation cephalosporins and were tested for the ESBL phenotype. The prevalence of ESBL-ECs recovered from retail meat samples is shown in Table 1. A total of 29 phenotypically positive ESBL-EC isolates were recovered from domestic (*n* = 18) and imported (*n* = 11) meat samples. The occurrences of ESBL-ECs in domestic pork and chicken meat were 0.2% (95% CI 0.0–1.0%) and 3.0% (95% CI 1.8–4.8%), respectively. There were no ESBL-EC isolates in the domestic beef samples. Meanwhile, the prevalence of ESBL-ECs in the imported beef, pork, and chicken meat were 0.1% (95% CI 0.0–1.0%), 0.5% (95% CI 0.1–1.6%), and 2.7% (95% CI 1.1–5.5%), respectively. The prevalence of ESBL-ECs in chicken meat was significantly higher than that in other meat samples (*p* < 0.001). No significant difference in the prevalence of ESBL-ECs was present between domestic and imported meat samples (*p* > 0.05).

### 3.2. AMR of ESBL-ECs

The AMR prevalence and profiles of phenotypically positive ESBL-ECs are shown in Table 2 and Appendix A. All isolates were resistant to AMP and CTF, whereas they were susceptible to AmC, FOX, and MEM. The most common non-beta-lactam resistance was present against NAL (75.9%, 22/29) and TET (72.4%, 21/29). All isolates showed the multidrug resistance phenotype, which means that the bacteria were resistant to three or more antimicrobial agents belonging to different categories. Meanwhile, no significant difference in the occurrence of resistance to each antimicrobial agent was present between isolates from domestic and imported meats (*p* > 0.05).

### 3.3. Distribution of Beta-Lactamase Genes and Plasmid Incompatibility Groups

The distribution of ESBL genes among 29 ESBL-EC isolates is shown in Figure 1. Out of the 29 ESBL-EC isolates, 28 carried *bla*_CTX-M_. One isolate (EC2017_203) not harboring *bla*_CTX-M_ had *bla*_TEM-1b_ and *bla*_SHV-12_. Further, 12 isolates harbored the combination of *bla*_CTX-M_ and *bla*_TEM_. The CTX-M genotypes in our ESBL-EC isolates were diverse, including *bla*_CTX-M-1_(*n* = 2), *bla*_CTX-M-2_(*n* = 2), *bla*_CTX-M-3_(*n* = 1), *bla*_CTX-M-8_(*n* = 3), *bla*_CTX-M-14_(*n* = 2), *bla*_CTX-M-15_(*n* = 5), *bla*_CTX-M-27_(*n* = 1), *bla*_CTX-M-55_(*n* = 11), and *bla*_CTX-M-65_(*n* = 2). Of these genotypes, the main type was *bla*_CTX-M-55,_ which was identified in 11 isolates (eight isolates from domestic chicken, two from imported chicken, and one from imported beef). The beta-lactamase genes of CTX-M-2 and CTX-M-8 group were not identified in ESBL-EC isolates from domestic meat. In addition to beta-lactamase genes, all strains carried genes conferring resistance to other classes of antimicrobial agents. Meanwhile, significant difference in the occurrence of each resistance gene was not present between isolates from domestic and imported meat (*p* > 0.05).

A total of 17 plasmid incompatibility types were identified in our collection of isolates using PlasmidFinder [48]. The most common plasmid replicon groups across all 29 ESBL-EC isolates were IncFⅠB (*n* = 23), followed by IncFⅡ (*n* = 11) and IncⅠ1 (*n* = 11). Meanwhile, the difference in the occurrence of plasmid incompatibility groups across ESBL-EC isolates from domestic meat and imported meat was not significant, except for IncFⅠB group (Appendix A).

### 3.4. STs of ESBL-EC

The results from MLST showed that the ESBL-ECs belonged to 21 different STs (Figure 2). The most frequent clonal types (STs) were ST58 (*n* = 3) belonging to phylogroup B1 and ST93 (*n* = 3) belonging to phylogroup A. Other STs were identified in less than two isolates. This implies that the STs of ESBL-EC isolates in this study were highly diverse.

### 3.5. Pan-Genome, Population Structure and Phylogeny of ESBL-EC

An average of 4933 genes per isolate was identified by automated annotation. An overall pan-genome consisted of total 14,094 genes, which made up the core gene set (including soft core gene) and the accessory gene set, comprising 3205 genes and 10,889 genes, respectively. The cumulative number of genes in the pan-genome continued to increase as more genomes were added to the collection of analysis (Appendix A). Our estimated pan-genome curve formula was: $y=2279.42x^{0.48}+2666.41$, where R^2^ was 0.9984. This result suggested that our ESBL-EC population have an open pan-genome, since a pan-genome is considered open when 0 < γ<1 [39,40].

A maximum likelihood phylogenetic tree of the 29 ESBL-ECs was constructed based on 188,735 SNPs in the core gene alignment. The phylogeny of core gene SNP revealed diverse clonal-related groups of ESBL-ECs, belonging to four major lineages, which generally correlated with the *E. coli* clonal ST. Meanwhile, three isolates belonging to ST93 (domestic chicken from different companies), two isolates belonging to ST2170 (domestic chicken from different companies), and two isolates belonging to ST602 (domestic chicken from different companies) were closely related, presenting average pairwise SNP differences of 67, 22, and 40, respectively.

A population structure analysis using the hierBAPS sequence clustering approach, based on SNP alignment of core-genomes [45], clustered 29 ESBL-ECs into four lineages: BAPS cluster 1 to BAPS cluster 4 (Figure 2). The BAPS clusters correlated with the *E. coli* phylogroups. All isolates from the phylogroup A (*n* = 6) and phylogroup B2 (*n* = 2) were assigned to the BAPS cluster 2 and BAPS cluster 4, respectively. The phylogroup B1 (*n* = 14) and C (*n* = 1) were included in BAPS cluster 1. The phylogroup F (*n* = 5) and phylogroup D (*n* = 1) isolates were included in BAPS cluster 3. Meanwhile, the BAPS clusters showed significant differences (*p* < 0.001) in the distribution of pairwise SNP differences in each cluster (Figure 3). The BAPS cluster 4 showed the lowest within-BAPS-cluster SNP diversity, characterized by lower average pairwise SNP differences of 2061. In comparison, BAPS cluster 1, 2, and 3 revealed higher average pairwise SNP differences of 16057, 17262, and 50465, respectively. These clusters also comprised a diverse set of STs: 10 STs in the BAPS cluster 1, 4 STs in the BAPS cluster 2, 6 STs in the BAPS cluster 3. Meanwhile there were no clusters of isolates specific to either source or their country groups, except for the BAPS cluster 4, which comprised Korean chicken meat isolates from the same company (phylogroup B2).

## 4. Discussion

In our study, we detected 29 ESBL-ECs from raw meat samples purchased at Korean retail stores between 2015 and 2018. Among the 29 ESBL-ECs, 18 isolates were recovered from domestic meat samples and 11 from imported meat samples. The observed 1.0% (18/1737) prevalence of ESBL-ECs in domestic meat samples was lower than that reported in a few previous Korean studies [24,26,27]. The observed 0.7% (11/1497) prevalence of ESBL-ECs from imported meat samples, however, was comparable with the previously reported prevalence of 1.1% (20/1771) in imported meat samples in Korea (*p* = 0.328) [31]. Meanwhile, the prevalence of ESBL-ECs in chicken was higher than that in pork and beef. This result was consistent with that of the previous studies regarding ESBL-EC prevalence in livestock in Korea [24,26,52], and in Germany, the United Kingdom, Turkey, and Spain [53,54,55,56].

All of the ESBL-EC isolates showed susceptibility of FOX (cephamycin) and MEM (carbapenem), as typical characteristics of ESBL-producing *Enterobacteriaceae*. All of the 1205 *E. coli* isolates were also susceptible to MEM (data not shown). Carbapenem-resistant *E. coli* have not yet been reported in food-producing animals and meat products, whereas carbapenem-resistant *E. coli* have been identified from human samples and companion animals in previous studies in Korea [23,57,58]. All of ESBL-ECs in this study also showed the multidrug resistance phenotype. This result is concordant with that of a previous study, which reported the high prevalence of multidrug resistant ESBL-ECs in raw chicken meat in Korea [30]. Meanwhile, the resistances against TET (tetracycline) and NAL (quinolone) were relatively high, suggesting that the common overuse of tetracyclines and quinolones in livestock farming worldwide has contributed to the acquisition of TET and NAL resistance [59].

In this study, 28 of 29 ESBL-EC isolates had CTX-M genotypes, just as CTX-M ESBLs have increased in occurrence worldwide since the 2000s [60]. Although genotypes of our ESBL-ECs were diverse, CTX-M group 1 (*bla*_CTX-M-1_, *bla*_CTX-M-3_, *bla*_CTX-M-15_, *bla*_CTX-M-55_) and CTX-M group 9 (*bla*_CTX-M-14_, *bla*_CTX-M-27_, *bla*_CTX-M-65_) were prevalent types, which was consistent with the findings of previous studies in Korea [25,26,27,28,30,32,52,61,62]. The most predominant CTX-M genotype was CTX-M-55, which is a variant of globally emerging CTX-M-15, and it varies by a single amino acid substitution (Ala77→Val) that contributes to enhancing the enzymatic activity [60,63]. The detection of *bla*_CTX-M-55_ has been growing in humans, animals, and food in Asia [29,64,65,66,67,68]. Genotypes of CTX-M-2 and CTX-M-8 were only identified in ESBL-EC isolates from imported meat samples, implying that the meat produced in foreign countries might be contaminated with ESBL-ECs during production in their countries. Most of our ESBL-EC isolates carried IncF plasmids, which have been reported to play a role in the dissemination of CTX-M-related enzyme genes [69]. However, we were unable to identify plasmid types that harbored CTX-M-related genes, because *bla*_CTX-M_ could not be assembled as one contig, along with replicon sequences.

The pan-genome analysis showed that our ESBL-EC isolates had an expanding pan-genome and this result is consistent with those of previous studies in pan-genome analyses of *E. coli* [70,71]. An open pan-genomic trait of *E. coli* probably contributes to the diversity of species [39]. The 29 ESBL-EC isolates were clustered into four lineages by BAPS, which correlated with *E. coli* phylogroups. This result was concordant with those of previous studies, which reported a correlation between BAPS clusters and *E. coli* phylogroups [70,72]. The BAPS cluster of interest in our study is the BAPS cluster 4, which includes the isolates obtained from domestic chicken samples. These isolates belong to ST95. Two isolates of ST95 belong to the B2 phylogroup, which includes most extraintestinal pathogenic *E. coli* (ExPEC) strains that cause infections [73,74]. ST95 strains have been reported to cause human infection and avian pathologies [74]. ST95 strains have also been reported to frequently appear as causative agents of urinary tract infections and blood sepsis in humans. Moreover, they have been consistently identified in poultry samples, suggesting that there may be a poultry reservoir of human ExPEC ST95 lineage [75]. In our study, two ST95 strains were isolated from cut chicken, processed by the same company on different dates. Further investigation is needed to identify the source of this contamination and to implement measures for constraining the spread of ST95. Other prevalent clonal groups (ST93, ST602, and ST2170) in domestic meat samples were isolated from chicken meat; these groups were isolated from chicken samples procured from different companies. Of these STs, ST93 and ST602 were reported to be infrequent in humans but more frequent in animal samples (specifically chicken) [24,76,77], suggesting that the strains of groups ST93 and ST602 were present in chicken, since farming stages and chicken act as reservoirs for these clonal groups. ST2170 was previously reported in retail chicken meat and turkey in Japan and Ecuador, respectively [68,78].

The 29 ESBL-EC isolates showed diverse clonal types, representing 21 different STs. The clonal group of ST58, one of the most frequent clonal types in this study, has been reported as a major vector of worldwide dissemination of ESBL-related genes along with ST155 [79,80]. All ST58 strains in our study were isolated from meat samples imported from different countries. ST58 of *E. coli* was not a common type in Korean human and animal samples [81], supporting the idea that ST58 strains from imported meat samples may be contaminated in the source countries. Although some pairs of isolates with the same clonal type were closely related by no more than 100 SNPs, the core-genome-based phylogeny of ESBL-EC isolates has revealed high diversity, which correlates with the diversity of clonal STs. This result suggests that ESBL-EC isolates have evolved from different ancestors and are not distributed through the clonal spread of predominant types.

A limitation of this study is the sampling design of ESBL-EC isolates. Due to the fact that just one *E. coli* isolate per meat sample was selected for ESBL phenotype confirmation, the prevalence of ESBL-ECs in our study could be underestimated. Moreover, we were unable to assemble the ESBL-gene-carrying plasmid sequences to further investigate the plasmid sequence diversity and compare plasmid distribution among ESBL-ECs isolated from different sources. Despite these limitations, this study provides a comprehensive overview of ESBL-EC diversity in retail meat in Korea. In addition, it provides information on the meat category that poses a high risk of spreading ESBL-ECs; this information can be used for containing the spread of ESBL-ECs.

In summary, we have described the prevalence and clonal diversity of ESBL-ECs isolated from retail meat samples in Korea. Our data showed that phylogenetically diverse ESBL-ECs with multidrug resistance phenotype were distributed nationwide, although the prevalence of ESBL-ECs in retail meat was 0.9%. The prevalence of ESBL-ECs in chicken (2.9%) was the highest when compared with that of other meat categories. Furthermore, ST95 strains, acting as pathogenic agents, were identified in chicken. Thus, as retail chicken may act as a potential vehicle for the spread of ESBL-ECs, including pathogenic types, it poses a health risk to consumers and food-handlers if contaminated with ESBL-producers. Therefore, close surveillance of ESBL-ECs should be continued, in order to establish a containment strategy for preventing the dissemination of ESBL-producers. Moreover, further studies will need to reduce the contamination of ESBL-producers throughout the food supply chain.

## Figures and Tables

**Figure 1 microorganisms-08-00508-f001:**
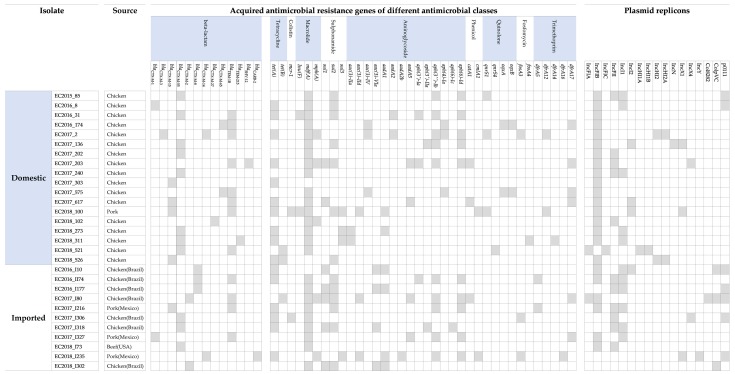
Distribution of acquired AMR (antimicrobial resistance) genes and plasmid incompatibility groups in ESBL-EC isolates. The panels are columns that represent presence or absence of AMR genes and plasmid replicons. The grey color indicates the presence of an AMR gene and plasmid replicon based on ResFinder [49] and PlasmidFinder [48], respectively.

**Figure 2 microorganisms-08-00508-f002:**
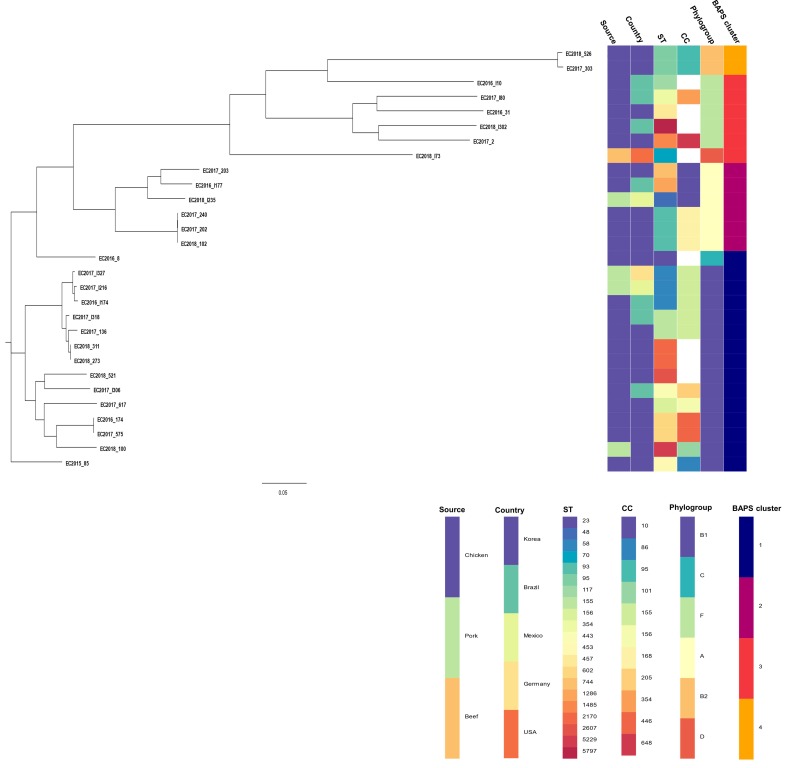
Maximum-likelihood core-genome SNP (single-nucleotide polymorphism) phylogenetic tree of ESBL-ECs isolated from retail raw meat in Korea with metadata (source and county). Phylogenetic tree was constructed based on 188,735 SNPs in the core-genome of 29 ESBL-EC isolates. MLST (multilocus sequence typing) sequence types (STs), clonal complex (CC), and phylogroups were identified using ClermonTyping [47], and sequence cluster was determined with hierBAPS (Bayesian analysis of population structure) [45]. Tree scale in the number of substitutions per site.

**Figure 3 microorganisms-08-00508-f003:**
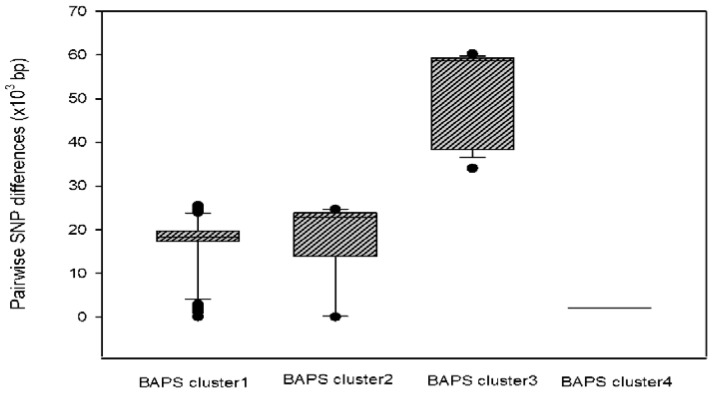
Distribution of pairwise SNP differences in each of the four BAPS clusters, to demonstrate the variations in sequence diversity in each cluster (*p* < 0.001).

**Table 1 microorganisms-08-00508-t001:** Prevalence of ESBL-EC (extended-spectrum beta-lactamase-producing *Escherichia coli*) isolates in retail meat samples.

Category	ESBL-EC Positive Rate, % (No. of ESBL-EC Confirmed Samples/ No. of Tested Samples)			
	Domestic	Imported	Total	*p*-Value **
Beef	0.0 (0/612)	0.1 (1/678)	0.1 (1/ 1290)	1.0000
Pork	0.2 (1/565)	0.5 (3/561)	0.4 (4/1126)	0.6114
Chicken	3.0 (17/560) *	2.7 (7/258) *	2.9 (24/818) *	0.9752
Total	1.0 (18/1737)	0.7 (11/1497)	0.9 (29/3234)	0.4736

* *p* < 0.001, difference between the proportions of each meat category by Chi-squared test. ** *p*-value, difference between the proportions of domestic and imported meat by Chi-squared test.

**Table 2 microorganisms-08-00508-t002:** AMR prevalence of the 29 ESBL-ECs isolated from retail meat samples.

Antibiotics		Resistance Rate, % (No. of Resistant Strains/ No. of Tested ESBL-ECs)			
		Domestic	Imported	Total	*p*-Value *
Beta-lactams	AmC	0.0 (0/18)	0.0 (0/11)	0.0 (0/29)	NA
	AMP	100.0 (18/18)	100.0 (11/11)	100.0 (29/29)	1.0000
	CAZ	22.2 (4/18)	0.0 (0/11)	13.8 (4/29)	0.2589
	CTF	100.0 (18/18)	100.0 (11/11)	100.0 (29/29)	1.0000
	FEP	16.7 (3/18)	9.1 (1/11)	13.8 (4/29)	0.9847
	FOX	0.0 (0/18)	0.0 (0/11)	0.0 (0/29)	NA
	MEM	0.0 (0/18)	0.0 (0/11)	0.0 (0/29)	NA
Non-beta-lactams	CHL	50.0 (9/18)	36.4 (4/11)	44.8 (13/29)	0.7401
	CIP	50.0 (9/18)	45.5 (5/11)	48.3 (14/29)	1.0000
	COL	11.1 (2/18)	18.2 (2/11)	13.8 (4/29)	1.0000
	GEN	38.9 (7/18)	45.5 (5/11)	41.4 (12/29)	1.0000
	NAL	83.3 (15/18)	63.6 (7/11)	75.9 (22/29)	0.4499
	STR	55.6 (10/18)	63.6 (7/11)	58.6 (17/29)	0.9679
	SXT	38.9 (7/18)	54.5 (6/11)	44.8 (13/29)	0.6615
	TET	77.8 (14/18)	63.6( 7/11)	72.4 (21/29)	0.6902
	MDR	100.0 (18/18)	100.0 (11/11)	100.0 (29/29)	1.0000

* *p*-value, difference between the proportions of domestic and imported meat by Chi-squared test. ** MDR, multidrug resistance; NA, not available.

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
