# Peer review of "Whole-Genome Sequencing-Based Characteristics in Extended-Spectrum Beta-Lactamase-Producing Escherichia coli Isolated from Retail Meats in Korea"

_microorganisms, 2020, doi:10.3390/microorganisms8040508_

Round 1

Reviewer 1 Report

SPECIFIC COMMENTS

  1. Abstract
  • Lines 17-21: No data regarding the number of characterized strains have been provided. The total number of coli isolates (3,234), the resistant to third-generation cephalosporins (1,197) and the number of ESBL-EC subjected to whole-genome sequencing (29) should be reported for a better understanding of the dimension of work.
  • Line 25: Include the % of prevalence (0.9%).

  1. Material and Methods.
  • Sample collection (page 2, lines 25-26): Can you provide more information regarding the surveillance program?: number of visited stores, locations, transport conditions of samples to the laboratory, etc.. Additional information regarding storing conditions of isolated strains for further characterization should be included.
  • Antimicrobial susceptibility testing (page 2, line 48). The cited reference (36) is older than the previous one (33). You must check and reorganize the references in accordance.

  1. Results.
  • Prevalence of ESBL-EC from retail raw meat samples
    • Page 4, line 11: You should clarify the number of coli strains processed in the study. According to the sentence, 3,234 E. coli were isolated (one per meat sample) but only 1,197 were tested for ESBL phenotype (provide a more clear sentence: E. coli was present in the 100% of tested samples and from the total of 3,234 isolates, 1,197 were resistant to the third-generation cephalosporins and tested for ESBL phenotype).
    • Page 4, line 19: The mentioned P-value (P<0.0001) is different to that showed in Table 1 (P<0.001).
  • AMR of ESBL-ECs
    • Page 4, line 28: The abbreviation of Meropenem used in the text is MERO, while MEM is included in the Table S1. Unify the terminology across the complete manuscript (discussion section included).
    • Page 6, line 6: Check the references included in the legend of Figure 1 (45 and 44). According to the References list should be 49 and 48.
  • STs of ESBL-EC
    • Page 7, lines 16-17: Check the references included in the legend of Figure 2 (43 and 41). According to the References list should be 45 and 47.
  • Pan-genome study
    • Page 8, lines 12-14: The meat source of reported closely related isolates should be included. If this information is considered relevant, additional comments in the discussion section should be appreciated.

  1. Discussion.

Despite a low prevalence of ESBL-EC multidrug resistant strains has been determined at retail meats in Korea, the final message is the potential risk that this fact poses to the consumers. Could you provided some discussion regarding the most risky meat in terms of prevalence, BAPS clusters or clonal types detected?

  1. Orthographic mistakes: check the following parts of the document
  • Page 1, line 13: Escherichia coli (italics of the final “i”).
  • Page 2, line 4: include and space before [14, 15].
  • Page 2, line 15: there is an extra space before “A number of studies…”.
  • Page 8, line 2: there is an extra space before “An overall..”.
  • References: the name of the Journals sometimes appears as the complete version and sometimes as the abbreviated one (capital letters or not). Check the complete list to unify criteria according to the Journal rules. In addition:
    • Reference 1: check the size of letter in line 2.
    • Reference 6: One Health (instead of one health).
    • Reference 7: World Health Organizaton (instead of Organization, W.H).
    • Reference 29: blaCTX-M-55 (instead of blaCTX-M-55).
    • Reference 41: Piscirickettsia salmonis (in italics).
    • References 48 and 51: The title of Journal should be before year of publication.
  • Figure S2: the number of genes should be 14,094 (instead of 14 094).
  • Table S1: MIC range tested concentration (instead of MIC range). Regarding the list of References:
    • Check the font of letter
    • Reference 2: Delete the dot before “Available…”

Author Response

We appreciate the reviewer's comments and suggestions. Please read our point-by-point responses below.

Reviewer 1

Point 1:

Abstract

  • Lines 17-21: No data regarding the number of characterized strains have been provided. The total number of coli isolates (3,234), the resistant to third-generation cephalosporins (1,197) and the number of ESBL-EC subjected to whole-genome sequencing (29) should be reported for a better understanding of the dimension of work.
  • Line 25: Include the % of prevalence (0.9%).

 Response 1: As per your suggestion, we have added the data regarding the number of characterized strains and the prevalence of ESBL-ECs (lines 17–26 in the revised version). Please note that we have corrected the number of E. coli strains from 1,197 to 1,205. We apologize for this error being present in the draft manuscript.

Point 2:

Material and Methods.

  • Sample collection (page 2, lines 25-26): Can you provide more information regarding the surveillance program?: number of visited stores, locations, transport conditions of samples to the laboratory, etc.. Additional information regarding storing conditions of isolated strains for further characterization should be included.
  • Antimicrobial susceptibility testing (page 2, line 48). The cited reference (36) is older than the previous one (33). You must check and reorganize the references in accordance.

 Response 2: As suggested by you, we have included more information regarding the surveillance program (lines 70–86 in the revised version). In our study, we isolated 1,205 E. coli strains from 3,234 raw meat samples, and out of these, 120 E. coli strains were resistant to third-generation cephalosporins. Further, 29 E. coli strains were confirmed to present the ESBL phenotype. To clarify more this, we have added the number of isolates and the number of third-generation cephalosporin-resistant strains (lines 88–99 in the revised version). According to your comment, we have also corrected the reference citation (lines 97 and 102 in the revised version). Thank you.

Point 3:

Results.

  • Prevalence of ESBL-EC from retail raw meat samples
    • Page 4, line 11: You should clarify the number of coli strains processed in the study. According to the sentence, 3,234 coli were isolated (one per meat sample) but only 1,197 were tested for ESBL phenotype (provide a more clear sentence: E. coli was present in the 100% of tested samples and from the total of 3,234 isolates, 1,197 were resistant to the third-generation cephalosporins and tested for ESBL phenotype).
    • Page 4, line 19: The mentioned P-value (P<0.0001) is different to that showed in Table 1 (P<0.001).
  • AMR of ESBL-ECs
    • Page 4, line 28: The abbreviation of Meropenem used in the text is MERO, while MEM is included in the Table S1. Unify the terminology across the complete manuscript (discussion section included).
    • Page 6, line 6: Check the references included in the legend of Figure 1 (45 and 44). According to the References list should be 49 and 48.
  • STs of ESBL-EC
    • Page 7, lines 16-17: Check the references included in the legend of Figure 2 (43 and 41). According to the References list should be 45 and 47.

 Response 3: According to your suggestion, we have corrected the sentences to provide clearer information on the number of tested meat samples, tested isolates, and tested strains for confirming the ESBL phenotype (lines 169–173 in the revised version). Further, according to your suggestion, we have unified the abbreviation of meropenem (MEM) throughout the manuscript and corrected the reference citation in Figure 1 and Figure 2. Thank you.

Point 4:

Results.

  • Pan-genome study
    • Page 8, lines 12-14: The meat source of reported closely related isolates should be included. If this information is considered relevant, additional comments in the discussion section should be appreciated.

 Response 4: As per your suggestion, we have included the data regarding the meat source (lines 248–250 in the revised version). And we have added some content in the discussion section (lines 311–322 in the revised version).

Point 5:

Discussion.

Despite a low prevalence of ESBL-EC multidrug resistant strains has been determined at retail meats in Korea, the final message is the potential risk that this fact poses to the consumers. Could you provide some discussion regarding the most risky meat in terms of prevalence, BAPS clusters or clonal types detected?

 Response 5: According to your suggestion, we have added some discussion regarding which meat category has high risk for consumers. The concerned sentence has been corrected to convey this with better clarity (lines 346–350 in the revised version).

Point 6:

Orthographic mistakes: check the following parts of the document

  • Page 1, line 13: Escherichia coli (italics of the final “i”).
  • Page 2, line 4: include and space before [14, 15].
  • Page 2, line 15: there is an extra space before “A number of studies…”.
  • Page 8, line 2: there is an extra space before “An overall..”.
  • References: the name of the Journals sometimes appears as the complete version and sometimes as the abbreviated one (capital letters or not). Check the complete list to unify criteria according to the Journal rules. In addition:
    • Reference 1: check the size of letter in line 2.
    • Reference 6: One Health (instead of one health).
    • Reference 7: World Health Organizaton (instead of Organization, W.H).
    • Reference 29: blaCTX-M-55 (instead of blaCTX-M-55).
    • Reference 41: Piscirickettsia salmonis (in italics).
    • References 48 and 51: The title of Journal should be before year of publication.
  • Figure S2: the number of genes should be 14,094 (instead of 14 094).
  • Table S1: MIC range tested concentration (instead of MIC range). Regarding the list of References:
    • Check the font of letter
    • Reference 2: Delete the dot before “Available…”

 Response 6: As per your suggestion, we have corrected these errors in the revised manuscript. Thank you.

Reviewer 2 Report

Study aims to provide knowledge on the prevalence of ESBL-ECs within retail meat in Korea.

I find the numbering per page to be an annoyance when reviewing having to details the page number and line number. Just make it continuous.

Limitations of one strain per meat sample? (Pg 2line 32). How many colonies were observed per twenty-five grams of each meat sample? Could there be heterogeneity in E. coli strains per meat sample? The authors should address the limitations of picking just one colony form a plate to test for ESBL phenotype. There is no mention of how many this 1 colony may have come from and thus may not be a true representative of the E coli strains present overall (e.g if picked from a plate containing 100 or more colonies).

Pg 3 line 7 (St. Petersburg State University, St. Petersburg, Russia) Probably not the correct reference for SPAdes.

Pg 3 line 10. Authors state the use of Roary and state that they have changed the percentage of isolates a gene must be in to be core from default [99] to 95%. However, they have not mentioned other key parameters such as minimum percentage identity for blastp which by default is 95%. Has anything else been changed?

Pg 3 Lines 26-29. Currently the BioProject (and data) are unavailable. Please could this be viewed. NCBI offer the function to provide links for referees to view data prior to release. There is no mention of raw data (reads) being uploaded to the SRA. I believe access to raw data is integral to allow further dissemination of sampling information for additional studies. This should be amended.

Pg 3 Lines 38 - 46. One questions the use of PCR amplification for MLST profiling of the ESBL-EC isolates if WGS was available?

Pg 4 lines 9-21. Surely it would be important to mention the type of meat that was used with regards to its processing history. For example, minced meat has the potential to have higher numbers of microorganisms per weight if processed in unsanitary environments due to the mixing of external portions of the meat, when compared with taking a large piece of meat or even an entire carcass (in the case of chicken) in which only external regions are likely to have become contaminated during the processing/supply chain. I suggest that this is, at the very least, mentioned.

Figure 1, Text resolution within the figure is not great. Further, the text is upside down.

Figure 2. Again. text upside down and resolution poor. I feel figure 2 is difficult to read with regards to the key detailing the colours looking like part of the digram. I don't feel that it lends itself towards easy reading. Trying to combined too much information into a single figure/tree can cause this problem.

Page 8 lines 6 to 9. It is accepted that E. coli has a diverse and open pangenome. I don't think this result is particularly novel considering the diverse STs detected or the small dataset size in reality. Further, there has been little attempt to assess the overall composition of the pangenome other than simply counting the number of coding sequences shared/not shared. I would think that some functional annotation of the pangenome would be important, further, exploration of genes involved in virulence and their distribution across the pangnome presented within this dataset is of interest/importance to understand how isolates maybe transferred through the food chain and into humans (if they do manage this).

Whilst the BAPS analysis is fairly useful, it simply reflects the topology of the tree, for example, the isolates in BAPS cluster 3 have the highest pairwise difference. This is obvious from just interpreting the tree itself. The authors state that tree scale is in substitutions per site, thus higher branch lengths between isolates mean higher number of substitutions between isolates. I think that whilst the authors are trying, to follow conventional methodologies, they haven't really questioned if the results add any additional understanding to what they have already produced from the data.

No mention of specialist vs generalists? Or how the ESBL-EC sequenced may cluster with E. coli hypothesised to be common in humans or poultry and whether they are specialists or generalists. This has implications on attributing the source of these E.coli with regards to the processing chain of the meat. Is it contamination from the rearing/farming environment or potentially from the processing of the meat with contamination coming from humans. This should be explored/discussed.

Author Response

We appreciate the reviewer's comments and suggestions. Please read our point-by-point responses below.

Point 1:

Study aims to provide knowledge on the prevalence of ESBL-ECs within retail meat in Korea.

I find the numbering per page to be an annoyance when reviewing having to details the page number and line number. Just make it continuous.

Limitations of one strain per meat sample? (Pg 2 line 32). How many colonies were observed per twenty-five grams of each meat sample? Could there be heterogeneity in E. coli strains per meat sample? The authors should address the limitations of picking just one colony form a plate to test for ESBL phenotype. There is no mention of how many this 1 colony may have come from and thus may not be a true representative of the E coli strains present overall (e.g if picked from a plate containing 100 or more colonies).

 Response 1: As you have correctly pointed out, this study does have a limitation regarding sampling design. Despite this limitation, our study provides comprehensive overview of the diversity of ESBL-producing E. coli in retail meat samples in Korea. It also provides information on the meat category that poses a high risk of spreading ESBL-producing E. coli, which can be used to prevent it. Considering this, we have added the following lines in the discussion section that elaborate on the limitations of our study (lines 335–342 in the revised version).

Point 2: Pg 3 line 7 (St. Petersburg State University, St. Petersburg, Russia) Probably not the correct reference for SPAdes.

Response 2: As per your suggestion, we have corrected this reference (line 109 in the revised version). Thank you.

Point 3: Pg 3 line 10. Authors state the use of Roary and state that they have changed the percentage of isolates a gene must be in to be core from default [99] to 95%. However, they have not mentioned other key parameters such as minimum percentage identity for blastp which by default is 95%. Has anything else been changed?

Response 3: We have used the default parameters of Roary to construct the core-genome of ESBL-producing E. coli. We have added the sentence regarding Roary parameters (lines 113–116 in the revised version). Please note that we corrected the sentences to discriminate core gene and soft core gene.

Point 4: Pg 3 Lines 26-29. Currently the BioProject (and data) are unavailable. Please could this be viewed. NCBI offer the function to provide links for referees to view data prior to release. There is no mention of raw data (reads) being uploaded to the SRA. I believe access to raw data is integral to allow further dissemination of sampling information for additional studies. This should be amended.

Response 4: Our whole-genome sequencing data were deposited at NCBI as assembled data. We requested if NCBI could provide the links for referees or reviewers to view data prior to release. However, we got answer that NCBI could not provide links for referees. Our genome data will be released when the paper regarding our sequence data is published. As suggested by you, we are considering uploading the raw data (reads). We have added information regarding the accession numbers and related strains (lines 132–142 in the revised version).

Point 5: Pg 3 Lines 38 - 46. One questions the use of PCR amplification for MLST profiling of the ESBL-EC isolates if WGS was available?

Response 5: We had already performed the conventional MLST for ESBL-EC isolates prior to their whole-genome sequencing.

Point 6: Pg 4 lines 9-21. Surely it would be important to mention the type of meat that was used with regards to its processing history. For example, minced meat has the potential to have higher numbers of microorganisms per weight if processed in unsanitary environments due to the mixing of external portions of the meat, when compared with taking a large piece of meat or even an entire carcass (in the case of chicken) in which only external regions are likely to have become contaminated during the processing/supply chain. I suggest that this is, at the very least, mentioned.

Response 6: As suggested by you, we have mentioned the type of meat (lines 170–171 in the revised version).

Point 7: Figure 1, Text resolution within the figure is not great. Further, the text is upside down. Figure 2. Again. text upside down and resolution poor. I feel figure 2 is difficult to read with regards to the key detailing the colours looking like part of the digram. I don't feel that it lends itself towards easy reading. Trying to combined too much information into a single figure/tree can cause this problem.

Response 7: As per your suggestion, in the revised version of the manuscript, we have replaced Figures 1 and 2 with figures having improved resolution.

Point 8: Page 8 lines 6 to 9. It is accepted that E. coli has a diverse and open pangenome. I don't think this result is particularly novel considering the diverse STs detected or the small dataset size in reality. Further, there has been little attempt to assess the overall composition of the pangenome other than simply counting the number of coding sequences shared/not shared. I would think that some functional annotation of the pangenome would be important, further, exploration of genes involved in virulence and their distribution across the pangnome presented within this dataset is of interest/importance to understand how isolates maybe transferred through the food chain and into humans (if they do manage this).

Whilst the BAPS analysis is fairly useful, it simply reflects the topology of the tree, for example, the isolates in BAPS cluster 3 have the highest pairwise difference. This is obvious from just interpreting the tree itself. The authors state that tree scale is in substitutions per site, thus higher branch lengths between isolates mean higher number of substitutions between isolates. I think that whilst the authors are trying, to follow conventional methodologies, they haven't really questioned if the results add any additional understanding to what they have already produced from the data.

Response 8: Thank you for your suggestion. We will study the functional comparison of pangenome within this dataset as a separate study in the future. BAPS clustering provides useful information when it is difficult to specify the boundaries of separate lineages in the phylogenetic tree.

Point 9: No mention of specialist vs generalists? Or how the ESBL-EC sequenced may cluster with E. coli hypothesised to be common in humans or poultry and whether they are specialists or generalists. This has implications on attributing the source of these E.coli with regards to the processing chain of the meat. Is it contamination from the rearing/farming environment or potentially from the processing of the meat with contamination coming from humans. This should be explored/discussed.

Response 9: As per your suggestion, we have added some points regarding strains of interest and whether they are specialists or generalists (lines 311–322 and lines 326–329 in the revised version).

Round 2

Reviewer 2 Report

Manuscript has been reviewed. Issues that I identified in my original review have been addressed such as figures, a few references and considerations to the limitations of approach used.